# Multi-morbidity and blood pressure trajectories in hypertensive patients: A multiple landmark cohort study

Jenny Tran[1], Robyn Norton[2], Dexter Canoy[1,3,4,5], Jose Roberto Ayala Solares[1], Nathalie Conrad[1], Milad Nazarzadeh[1,3], Francesca Raimondi[1], Gholamreza Salimi-Khorshidi[1,3], Anthony Rodgers[2], Kazem Rahimi[1,3,4,6]*

1 Deep Medicine, Oxford Martin School, University of Oxford, Oxford, United Kingdom, 2 The George Institute for Global Health, University of New South Wales, Sydney, Australia, 3 Nuffield Department of Women's and Reproductive Health, University of Oxford, Oxford, United Kingdom, 4 National Institute of Health Research Oxford Biomedical Research Centre, Oxford, United Kingdom, 5 Faculty of Medicine, University of New South Wales, Sydney, Australia, 6 Oxford University Hospitals NHS Foundation Trust, Oxford, United Kingdom

* kazem.rahimi@wrh.ox.ac.uk

**Data Availability Statement:** The data underlying this study were provided with permission from the CPRD (www.cprd.com); similar data may be requested from them.

## Abstract

### Background

Our knowledge of how to better manage elevated blood pressure (BP) in the presence of comorbidities is limited, in part due to exclusion or underrepresentation of patients with multiple chronic conditions from major clinical trials. We aimed to investigate the burden and types of comorbidities in patients with hypertension and to assess how such comorbidities and other variables affect BP levels over time.

### Methods and findings

In this multiple landmark cohort study, we used linked electronic health records from the United Kingdom Clinical Practice Research Datalink (CPRD) to compare systolic blood pressure (SBP) levels in 295,487 patients (51% women) aged 61.5 (SD = 13.1) years with first recorded diagnosis of hypertension between 2000 and 2014, by type and numbers of major comorbidities, from at least 5 years before and up to 10 years after hypertension diagnosis. Time-updated multivariable linear regression analyses showed that the presence of more comorbidities was associated with lower SBP during follow-up. In hypertensive patients without comorbidities, mean SBP at diagnosis and at 10 years were 162.3 mm Hg (95% confidence interval [CI] 162.0 to 162.6) and 140.5 mm Hg (95% CI 140.4 to 140.6), respectively; in hypertensive patients with ≥5 comorbidities, these were 157.3 mm Hg (95% CI 156.9 to 157.6) and 136.8 mm Hg (95% 136.4 to 137.3), respectively. This inverse association between numbers of comorbidities and SBP was not specific to particular types of comorbidities, although associations were stronger in those with preexisting cardiovascular disease. Retrospective analysis of recorded SBP showed that the difference in mean SBP 5 years before diagnosis between those without and with ≥5 comorbidities was −9 mm Hg (95% CI −9.7 to −8.3), suggesting that mean recorded SBP already differed according to the

**Funding:** This research was funded by the National Institute for Health Research (NIHR) Oxford Biomedical Research Centre and the Oxford Martin School. The views expressed are those of the authors and not necessarily those of the National Health Service, the NIHR or the Department of Health and Social Care. KR is in receipt of grants from: National Institute for Health Research (NIHR) Oxford Biomedical Research Centre; the Oxford Martin School; the British Heart Foundation and UKRI's Global Challenge Research Fund, Grant Ref: ES/P011055/1. JT was supported by the Rhodes Trust and Clarendon Fund for this work. MN received support from the British Heart Foundation (grant number: FS/19/36/34346), outside of this work. NC is supported by a fellowship from the Marie Skłodowska-Curie Actions programme and a grant from the European Society of Cardiology, outside of this work. All funders did not play a role in the study design, data collection and analysis, decision to publish, or preparation of the manuscript.

**Competing interests:** I have read the journal's policy and the authors of this manuscript have the following competing interests: KR is a member of PLOS Medicine's Editorial Board. AR is employed by The George Institute for Global Health (TGI) and seconded part-time to George Medicines Pty Ltd (GM). TGI has submitted patent applications in respect of low fixed-dose combination products for the treatment of cardiovascular or cardiometabolic disease, and AR is listed as one of the inventors; George Health Enterprises Pty Ltd (GHE) and its subsidiary, GM, have received investment funds to develop fixed-dose combination products, including combinations of blood pressure-lowering drugs. GHE is the social enterprise arm of TGI. AR does not have a direct financial interest in these patent applications or investments.

**Abbreviations:** BMI, body mass index; BP, blood pressure; CI, confidence interval; CPRD, Clinical Practice Research Datalink; DBP, diastolic blood pressure; GP, general practice; QOF, Quality and Outcomes Framework; SBP, systolic blood pressure; STROBE, STrenghtening the Reporting of Observational studies in Epidemiology.

presence of comorbidity before baseline. Within 1 year after the diagnosis, SBP substantially declined, but subsequent SBP changes across comorbidity status were modest, with no evidence of a more rapid decline in those with more or specific types of comorbidities. We identified factors, such as prescriptions of antihypertensive drugs and frequency of healthcare visits, that can explain SBP differences according to numbers or types of comorbidities, but these factors only partly explained the recorded SBP differences. Nevertheless, some limitations have to be considered including the possibility that diagnosis of some conditions may not have been recorded, varying degrees of missing data inherent in analytical datasets extracted from routine health records, and greater measurement errors in clinical measurements taken in routine practices than those taken in well-controlled clinical study settings.

## Conclusions

BP levels at which patients were diagnosed with hypertension varied substantially according to the presence of comorbidities and were lowest in patients with multi-morbidity. Our findings suggest that this early selection bias of hypertension diagnosis at different BP levels was a key determinant of long-term differences in BP by comorbidity status. The lack of a more rapid decline in SBP in those with multi-morbidity provides some reassurance for BP treatment in these high-risk individuals.

## Author summary

### Why was this study done?

- Some have suggested that patients with multi-morbidity have lower blood pressure (BP) than those without, but these studies were mainly based on cross-sectional investigations and only considered a limited number of co-occurring chronic conditions.

- It has been reported that BP levels tend to fall in years preceding death, with steeper declines in patients with cardiovascular comorbidities, but underlying determinants of this pattern remain uncertain.

### What did the researchers do and find?

- In a large number of patients with newly diagnosed hypertension, we estimated mean BP for each year many years prior to and after the diagnosis, separately according to the number and type of comorbidities.

- Those with more additional comorbidities had lower mean systolic blood pressure (SBP) at the time of the diagnosis; this difference persisted up to 10 years from the time of diagnosis and was apparent even some years prior to the diagnosis.

- Although SBP varied by type of comorbidity, there was no single type that was associated with higher SBP during follow-up.

**What do these findings mean?**

- The BP level at the time of diagnosis of hypertension was the key determinant of long-term BP differences according to number or type of comorbidity.

- This selection bias of hypertension diagnosis at different BP levels needs to be considered when comparing BP of patients according to their comorbidity status.

- The lack of a more rapid decline in SBP in those with multi-morbidity provides some reassurance for BP treatment in these high-risk individuals.

## Introduction

Up to two-thirds of patients with hypertension have other comorbidities [1]. Yet, patients with multi-morbidity are commonly excluded from or underrepresented in major clinical trials, hence, limiting the evidence on how best to manage elevated blood pressure (BP) of hypertensive patients with additional comorbidities. With the expected rise in the burden of multi-morbidity as well as hypertension [2], it is important to understand how the presence of comorbidities affects BP to inform future policy and research.

Several studies have reported associations between individual comorbidities, such as ischaemic heart disease, heart failure, depression, or dementia, and BP and have mostly concluded that presence of such comorbidities was associated with lower BP [3–6]. However, most analyses have been cross-sectional and based on limited number of patients or types of comorbidities and could not take account of several confounding factors that could determine differences in BP. Some limited evidence from longitudinal studies have been reported, suggesting a generally declining BP in years preceding death, with this downward BP trajectory being steeper in people with cardiovascular comorbidities or dementia [7]. However, the underlying reasons for such observations remain unclear.

From a clinical point of view, it is important to understand whether certain comorbidities per se reduce BP over time which could justify the cautious guideline recommendations for BP treatment in such patients or whether associations are due to other factors such as age, body mass index (BMI), more intensive treatment in patients with multi-morbidity, or indeed differences in intensity of care and attention received.

We sought to investigate the association between comorbidity status and subsequent BP in a large cohort of patients with incident hypertension, leveraging information on timing of disease diagnosis, change in treatments, and consideration of other important determinants of BP levels over time.

## Methods

### Data source

The study was conducted using linked electronic health records from the UK Clinical Practice Research Datalink (CPRD) from its inception on January 1, 1985 to September 30, 2015 [8]. The CPRD database at the time of conducting this study included data from 674 general practices, covering approximately 7% of the UK population and broadly represented the population by age, sex, and ethnicity [9]. This database is linked to other national administrative databases including hospitalisations (Hospital Episode Statistics), death registration (Office of

National Statistics), and the Index of Multiple Deprivation [10–12], which makes CPRD database a comprehensive resource for prospective analysis of UK primary care data. The validity and reliability of recorded diagnoses for a range of major chronic conditions have been reported previously, reporting an average positive predictive value of 89%, with 92% completeness when compared to national registries [13,14]. The CPRD maintains an audit and determines practices providing clinical data of acceptable quality for research purposes. In this research, we only considered clinical records that have met research quality standards and were linked to hospitalisation and mortality databases. This study was conducted as part of an investigation into multi-morbidity and cardiovascular disease, and the research protocol was submitted to, and approved by, the CPRD Independent Scientific Advisory Committee (protocol number 16_049R). No additional informed consent was required.

## Study design

We used a multiple landmark cohort design in order to investigate associations prospectively with time-updated information that takes advantage of the dynamic nature of electronic health records (**S1 Fig**) [15]. In this design, the baseline was set at the time of first recorded diagnosis of hypertension, and "landmarks" were set yearly up to 5 years before and 10 years after the baseline date of hypertension diagnosis. The patient cohort within each yearly landmark time point were followed for at least 6 months, and relevant variables were considered during this period within the landmark time year.

## Study population

Patients aged 16 years and over with quality data were considered for inclusion based on previously described criteria, and the selection of study population is shown in **Fig 1** [2]. Patients with a diagnostic code for hypertension (**S1 Table**) recorded between January 1, 2000 and December 31, 2014 were considered for inclusion into the study. UK guidelines recommend a minimum of 2 consecutive BP measurements for diagnosis of hypertension. However, in primary care records, a single BP measurement is typically recorded from each consultation [7]. We confined the study population to those with newly diagnosed hypertension and excluded prevalent diagnoses, defined as any recording of hypertension within less than 12 months after the current registration date with their general practice [16]. We further excluded patients who died within 2 years of hypertension diagnosis, to reduce the risk of reverse causality. Patients were followed from the date of hypertension diagnosis until the earliest of date of death, transfer out of practice, last collection date of clinical data for the practice, or end of study on December 31, 2014.

## Study outcome

Our primary outcome was systolic blood pressure (SBP) as a continuous variable. Data capture for SBP was conducted for each landmark year using recorded SBP within 3 months of a landmark time point (**S1 Fig**). Within a landmark year, we assessed the exposure and determined the SBP recorded within 3 months after the determination of comorbidity status. In case of multiple BP readings during this time window, we took the mean of all recordings. If there were no BP readings during this period, nearest neighbour imputation was used with univariate time series modelling, which assumed autocorrelation between non-equidistant time intervals [17]. We applied multivariable regression models to estimate differences in SBP according to comorbidity status for each landmark year and used simulations for predicted values (based on Zelig package in R) to determine mean SBP for each category of number or type of comorbidity.

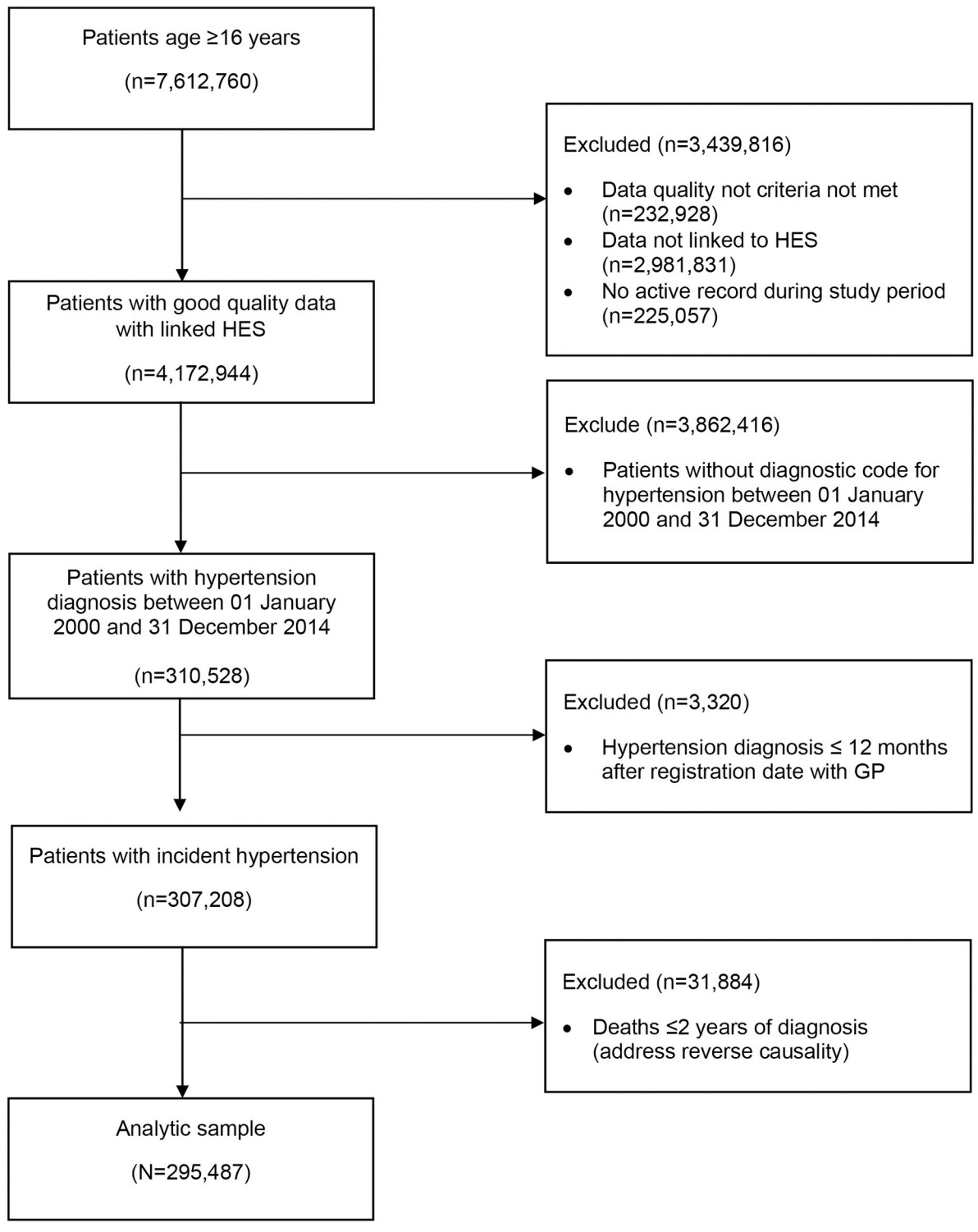

HES – Hospital Episode Statistics; GP – General Practice.

**Fig 1. Flowchart of selection of patients into the study.**

## Selection and definition of comorbidities and covariates

Our main exposure was comorbidity status classified as the number of comorbidities (hypertension alone as the reference group) as well as by the type of comorbidity or comorbidity group (no comorbidity group as the reference category). We selected 22 comorbidities that were considered clinically important and highly prevalent, containing both related and unrelated conditions to hypertension, categorised into 6 groups (cardiometabolic, mental and cognitive, respiratory, musculoskeletal, cancers, and haematological) (**S2 Table**). Our approach for selection of comorbidities has been described before [2]. In brief, selection was based on 3 sources: (1) Quality and Outcomes Framework (QOF), an incentive scheme for general practitioners in the UK [18]; (2) Charlson Comorbidity Index, the most commonly used comorbidity index originally designed to predict inpatient hospital mortality [19]; and (3) Multiple chronic conditions list, chosen by the United States Department of Health and Human Services for the initiative on multiple chronic conditions [20]. Comorbidity status was time-updated for each cohort and captured all available recorded diagnoses from CPRD inception to up to 6 months before the time window of outcome assessment to accurately capture a patient's total comorbidity burden before outcome measurement. Demographic factors, deprivation level (based on fifths of the Index of Multiple Deprivation), ethnicity (as recorded by the general practice [GP]), smoking status, BMI, total cholesterol, and antihypertensive treatment use (based on recorded prescriptions) were also extracted (**S1 Fig**) from up to 12 months before to time of hypertension diagnosis. To update factors with sufficient repeated time-varying information (e.g., antihypertensive treatment use), data were extracted within 6 to 18 months prior to exposure assessment at each landmark year (**S1 Fig**).

## Statistical analyses

Multivariable linear regression was used to model differences in SBP by comorbidity status, and the calculated values were used to derive mean SBP from 1,000 simulations by asymptotic normal approximation to the log-likelihood [21]. We included time-updated number of prescribed antihypertensives as a covariate into the models to assess the associations between comorbidity status and SBP independently of differences in antihypertensive prescription. Antihypertensive classes (5 classes) were defined by product codes under the British National Formulary codes (**S3 Table**).

All models were further adjusted for age (continuous variable), sex, deprivation level, ethnicity, smoking status, BMI (continuous variable), and year of diagnosis of hypertension. Age, comorbidities, number of prescribed antihypertensive classes, and SBP were time-updated for each landmark time point. By design, some patients who develop additional comorbidity may be classified to a different exposure group for that particular landmark year. Missing covariates for smoking and BMI were imputed using multiple imputation by expectation–maximisation with bootstrapping with 5 imputation datasets [22]. For each model, adjustments were made stepwise, and tests for collinearity, model fit, and interactions with age and sex were performed.

Moreover, we examined the association of comorbidity status with BP by (1) removing adjustment for antihypertensive medications to assess the contribution of BP treatment on the observed SBP differences; (2) additionally adjusting models for healthcare utilisation, using the frequency of BP measurements in the interval of 5 years before the hypertension diagnosis as a proxy for service utilisation; and (3) additionally adjusting models for SBP at diagnosis.

All statistical analyses were performed using R version 3.3.2. All our analyses have been pre-specified except for using BMI as continuous variable in the model which has been suggested by one of the reviewers. Reporting of this study was done in accordance with STrenghtening

the Reporting of Observational studies in Epidemiology (STROBE) guidelines [23] (**S1 STROBE Checklist**).

## Results

### Baseline characteristics

Between January 1, 2000 and December 31, 2014, there were 295,487 patients (51% women) aged 61.5 (SD = 13.1) years, on average, with incident hypertension. At the time of diagnosis (baseline), the mean SBP/diastolic blood pressure (DBP) were 159/91 mm Hg, and the mean total cholesterol was 5.5 mmol/L. A total of 96% were of white ethnicity, 14.8% in the most deprived fifth, 38.5% obese, 20.1% currently smoked, and 2.7% received prescriptions of ≥3 classes of antihypertensive drugs (**Table 1**). The proportions with missing values for some covariates were 59% for ethnicity, 41.7% for BMI, 25.4% for smoking status, and 35.4% for total cholesterol. On average, patients had 15.5 recorded measurements of BP after hypertension diagnosis and 8 measurements prior to the diagnosis (**S4 Table**).

Patients with ≥5 comorbidities, compared to those with hypertension alone, were more likely to be older, female, ex-smokers, and in the most deprived fifth of the population (**Table 1**). Patients with more comorbidities had lower mean SBP and DBP. The crude difference in SBP was 8.1 mm Hg between those with 5 or more comorbidities (mean SBP = 152.9 mm Hg) and those with hypertension alone (mean SBP = 161.0 mm Hg). Patients with more comorbidities were also prescribed more classes of antihypertensives compared to patients with hypertension alone. The proportion of patients with higher numbers of comorbidities increased over time after diagnosis (**S5 Table**). In the year of hypertension diagnosis, 36.1% had no comorbidities, and 3.2% had ≥5 comorbidities. These proportions changed to 10.7% and 18.6%, respectively, after 10 years of hypertension diagnosis. Patients with cardiometabolic comorbidities compared to those without cardiometabolic comorbidities were generally older (mean age 63.5 years versus 60.3 years), had lower SBP/DBP (156/88 mm Hg versus 161/93 mm Hg), and had lower total cholesterol (5.3 mmol/L versus 5.7 mmol/L) (**S6 Table**).

### Temporal analysis of associations between number of comorbidities and BP

Patients with higher numbers of comorbidities had a lower mean SBP at the time of diagnosis. Those with hypertension alone had an adjusted mean SBP of 162.3 mm Hg (95% confidence interval [CI] 162.0 to 162.6) compared to 157.3 mm Hg (95% CI 156.9 to 157.6) in those with ≥5 comorbidities. All patient groups followed a trend of decreasing SBP after diagnosis of hypertension. The largest decline in SBP was observed in the first year after diagnosis of hypertension with only modest changes afterwards, and parallel trends by number of comorbidities (**Fig 2**). Overall, patients with higher numbers of comorbidities maintained a lower SBP over time but with no evidence of a faster BP decline (**Fig 2**, **S2 Fig**). For instance, those with hypertension alone had an adjusted mean SBP of 145.2 mm Hg (95% CI 144.9 to 145.2) at 1 year after diagnosis and 140.5 mm Hg (95% CI 140.4 to 140.6) at 10 years after diagnosis; compared to those with ≥5 comorbidities, the corresponding SBP were 140.5 mm Hg (95% CI 140.2 to 140.8) and 136.8 mm Hg (95% CI 136.4 to 137.3), respectively. Retrospective analyses of SBP up to 5 years prior to the diagnosis of hypertension showed that the differences in adjusted mean SBP were even more pronounced in the years preceding than in the years after the diagnosis of hypertension (**Fig 2**). For example, at 5 years before diagnosis, the difference in mean adjusted SBP between those with ≥5 comorbidities versus those with hypertension alone was 9.0 mm Hg (95% CI 8.3 to 9.7), and the corresponding difference at 5 years after diagnosis was 4.4 mm Hg (95% CI 4.1 to 4.7). The pattern of lower mean SBP with more comorbidities remained similar in men and women (**S3** and **S4** **Figs**).

**Table 1. Baseline characteristics of incident hypertensive patients between 2000 and 2014 in the UK, by number of comorbidities.**

| Characteristic | All patients (n = 295,487) | Number of comorbidities | | | | | |
|---|---|---|---|---|---|---|---|
| | | 0 | 1 | 2 | 3 | 4 | 5 |
| | | (n = 106,801) | (n = 86,691) | (n = 52,432) | (n = 27,185) | (n = 12,824) | (n = 9,554) |
| **Age [years], mean (SD)** | 61.5 (13.1) | 58.9 (12.7) | 61.3 (13.0) | 63.0 (13.0) | 64.9 (12.8) | 66.5 (12.6) | 69.7 (12.0) |
| <65, % (n) | 60.7 (179,239) | 69.8 (74,502) | 61.4 (53,229) | 55.8 (29,240) | 49 (13,334) | 44.2 (5,667) | 34.2 (3,267) |
| ≥65, % (n) | 39.3 (116,248) | 30.2 (32,299) | 38.6 (33,462) | 44.2 (23,192) | 51.0 (13,851) | 55.8 (7,157) | 65.8 (6,287) |
| **Women, % (n)** | 50.7 (149,787) | 43.5 (46,415) | 50.5 (43,807) | 56.2 (29,480) | 59.7 (16,228) | 61.9 (7,935) | 62.0 (5,922) |
| **White ethnicity, % (n)** | 96.1 (116,574) | 95.8 (36,963) | 96.1 (34,115) | 96.4 (22,600) | 96.2 (12,192) | 96.6 (6,005) | 96.3 (4,699) |
| | [174,197] | [68,198] | [51,200] | [28,998] | [14,517] | [6,609] | [4,675] |
| **Fifths of deprivation index, % (n)** | | | | | | | |
| Q1 (least deprived) | 23.3 (68,902) | 25.6 (27,347) | 23.8 (20,607) | 21.8 (11,441) | 20.7 (5,615) | 18.1 (2,319) | 16.5 (1,573) |
| Q2 | 22.6 (66,699) | 23.6 (25,214) | 22.8 (19,745) | 21.9 (11,457) | 21.4 (5,807) | 20.4 (2,622) | 19.4 (1,854) |
| Q3 | 21.0 (62,142) | 20.8 (22,245) | 21.3 (18,430) | 21.3 (11,143) | 20.8 (5,664) | 20.8 (2,661) | 20.9 (1,999) |
| Q4 | 18.2 (53,722) | 17.4 (18,558) | 17.9 (15,551) | 18.8 (9,853) | 19.2 (5,222) | 20.2 (2,589) | 20.4 (1,949) |
| Q5 (most deprived) | 14.8 (43,718) | 12.5 (13,323) | 14.1 (12,259) | 16.2 (8,490) | 17.8 (4,852) | 20.5 (2,623) | 22.7 (2,171) |
| **Mean SBP (SD), mm Hg** | 159.1 (21.1) | 161.0 (21.4) | 159.2 (20.8) | 158.2 (20.6) | 156.6 (20.8) | 155.7 (21.0) | 152.9 (21.4) |
| **Mean DBP (SD), mm Hg** | 91.0 (12.5) | 93.4 (12.2) | 91.3 (12.1) | 89.8 (12.1) | 88.0 (12.3) | 86.6 (12.6) | 83.4 (12.8) |
| **Mean (SD) BMI, kg/m²** | 29.4 (6.1) | 28.7 (5.5) | 29.3 (5.9) | 29.7 (6.4) | 30.0 (6.6) | 30.3 (6.9) | 30.1 (6.9) |
| | [123,199] | [52,053] | [37,132] | [19,620] | [8,774] | [3,573] | [2,047] |
| **Smoking status, % (n)** | | | | | | | |
| Current | 20.1 (44,376) | 20.8 (15,826) | 19.7 (12,589) | 20.2 (8,134) | 19.2 (4,153) | 20.0 (2,119) | 18.8 (1,555) |
| Never | 47.7 (105,304) | 50.8 (38,608) | 48.9 (31,208) | 46.0 (18,511) | 44.2 (9,555) | 40.2 (4,261) | 38.1 (3,161) |
| Ex-smoker | 32.1 (70,871) | 28.4 (21,552) | 31.4 (20,015) | 33.8 (13,583) | 36.6 (7,919) | 39.9 (4,232) | 43.1 (3,570) |
| | [74,936] | [30,815] | [22,879] | [12,204] | [5,558] | [2,212] | [1,268] |
| **Mean total cholesterol (SD), mmol/L** | 5.5 (1.2) | 5.7 (1.1) | 5.6 (1.1) | 5.5 (1.2) | 5.3 (1.2) | 5.1 (1.2) | 4.9 (1.2) |
| | [104,581] | [44,923] | [31,425] | [16,844] | [7,104] | [2,864] | [1,421] |
| **Year of hypertension diagnosis, % (n)** | | | | | | | |
| 2000 | 5.0 (14,822) | 5.8 (6,195) | 5.2 (4,477) | 4.5 (2,383) | 3.8 (1,038) | 3.8 (490) | 2.5 (239) |
| 2014 | 3.4 (10,129) | 2.8 (2,985) | 3.2 (2,786) | 3.7 (1,951) | 4.4 (1,185) | 4.8 (613) | 6.4 (609) |
| **Number of antihypertensive classes prescribed, % (n)** | | | | | | | |
| 1 | 13.9 (41,137) | 9.0 (9,641) | 13.3 (11,508) | 17.0 (8,900) | 20.3 (5,522) | 23.5 (3,011) | 26.7 (2,555) |
| 2 | 6.5 (19,267) | 4.0 (4,238) | 5.3 (4,636) | 7.4 (3,875) | 10.4 (2,819) | 13.7 (1,751) | 20.4 (1,948) |
| ≥3 | 2.7 (8,058) | 1.4 (1,461) | 1.9 (1,611) | 3.0 (1,560) | 4.7 (1,283) | 6.8 (871) | 13.3 (1,272) |

The category percentages refer to complete cases. Numbers in square brackets refer to number of patients with missing data for the relevant characteristic. Deprivation level based on IMD 2015 cut by quintiles, where Q1 is least deprived fifth, and Q5 is most deprived fifth of the population. Number of antihypertensive classes refers to antihypertensives grouped into 5 classes: angiotensin converting enzyme inhibitor and angiotensin II receptor blocker, beta-blockers, calcium channel blockers, diuretics, and others (see Methods and S3 Table).

BMI, body mass index; DBP, diastolic blood pressure; IMD, Index of Multiple Deprivation; SBP, systolic blood pressure.

To further investigate the factors that might explain the association between the number of comorbidities and BP, we examined the impact of type of comorbidities, antihypertensive treatment use, and healthcare exposure on the association.

## Type of comorbidities

Analysis by type of comorbidity showed that adjusted mean SBP was lower across all groups of comorbidities. However, the magnitude of difference varied by comorbidity groups and individual comorbidities (Fig 3). The largest difference in adjusted SBP was seen in patients with

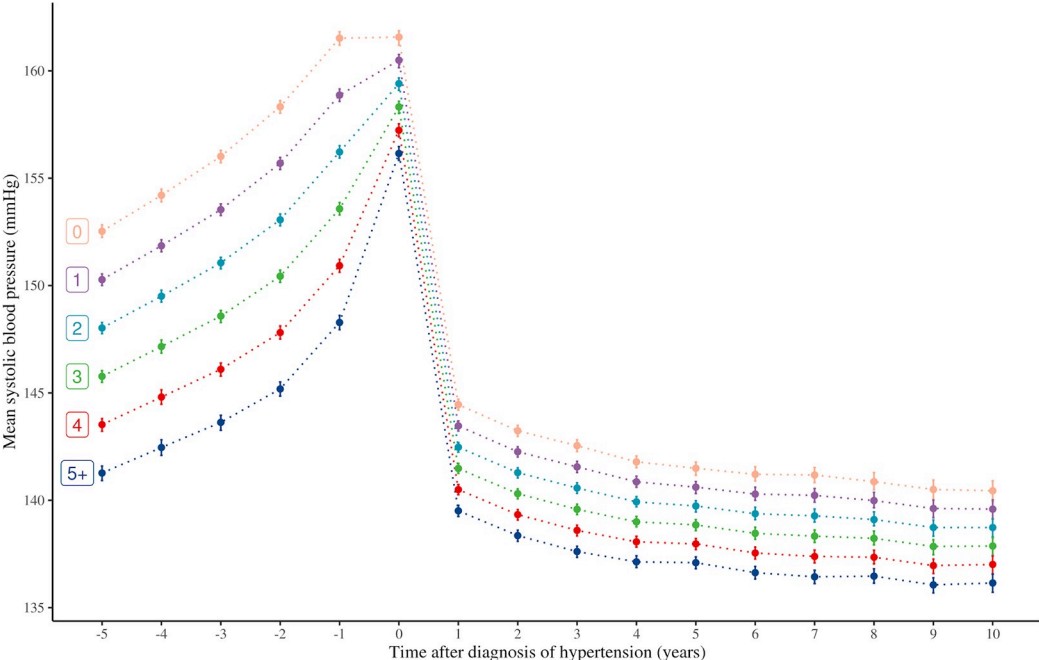

**Fig 2. Mean SBP over time by number of comorbidities.** SBP was calculated from linear regression models for each landmark cohort. Each line represents number of comorbidities in addition to hypertension: 0,1,2,3,4,5+. Bars for each dot represent 95% CIs. Negative time indicates time (year) before diagnosis of hypertension. Models were adjusted for age, sex, deprivation level, ethnicity, BMI, smoking status, number of classes of prescribed antihypertensive medications, and year of diagnosis of hypertension. BMI, body mass index; CI, confidence interval; SBP, systolic blood pressure.

cardiometabolic comorbidities compared to those with no such comorbidities; at 1 year after diagnosis of hypertension, patients with any cardiometabolic condition had a 2.4 mm Hg (95% CI 2.3 to 2.5) lower adjusted SBP than those without any cardiometabolic conditions. Results separately for men and women are shown in **S5 and S6 Figs**. The pattern of lower SBP in patients with cardiometabolic comorbidities, when compared to those without such comorbidities, was also seen when we retrospectively compared SBP in the years before diagnosis of hypertension. This difference was also larger in magnitude in years prior to, rather than at the time of, diagnosis of hypertension (**S2 Fig**).

## Antihypertensive treatment

Patients with more comorbidities were prescribed more classes of antihypertensives before and after hypertension diagnosis, although the relative difference in treatment intensity attenuated after diagnosis (**S6 Table**). We then explored the extent to which this differential treatment intensity could explain the pattern of BP according to comorbidity status. While adjusting for this variable slightly attenuated the relationship between number of comorbidities and SBP, the variation in SBP by comorbidity status persisted (**Fig 4**).

## Healthcare utilisation

To further explore the extent to which differences in SBP by comorbidity status might be due to differences in frequency of patient interactions with health services ("informed presence bias"), we measured the frequency of BP recording during the interval 5 years before and 5 years after hypertension diagnosis (**S4 Table**). Those with higher numbers of comorbidities had a higher frequency of BP measurements before the diagnosis of hypertension, compared

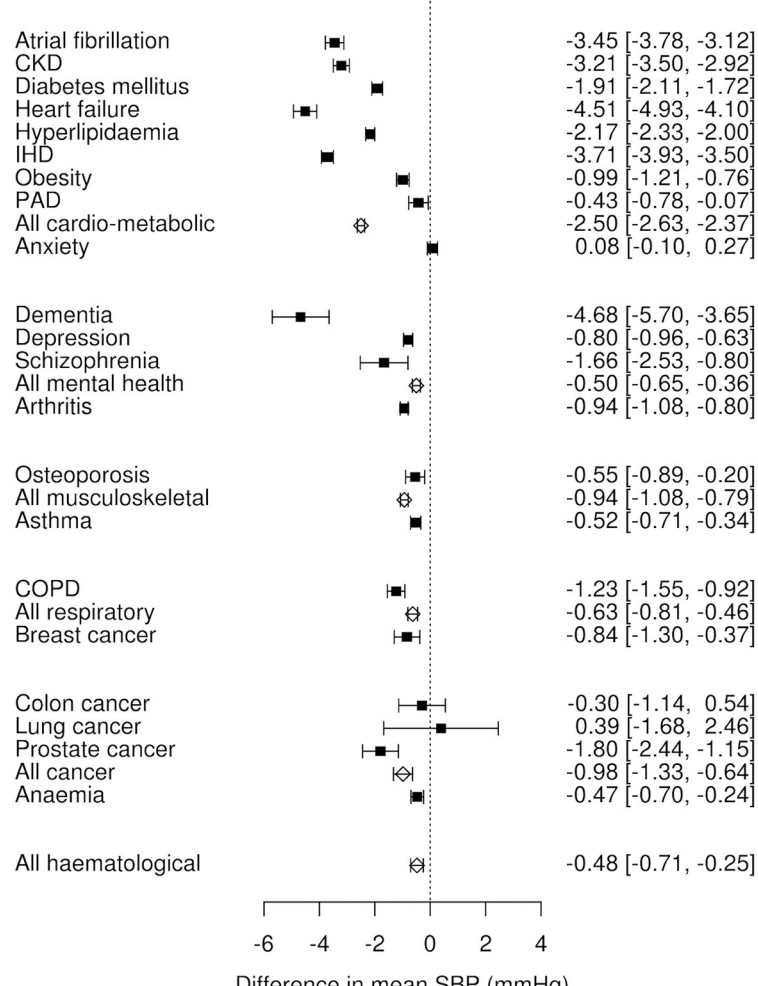

**Fig 3. Adjusted mean differences in SBP at 1 year after hypertension diagnosis stratified by comorbidity status.**
All models were adjusted for age, sex, deprivation level, ethnicity, cholesterol, BMI, smoking status, number of classes of prescribed antihypertensive medications, and year of diagnosis of hypertension. Reference group for each point estimate was patients without that particular comorbidity. BMI, body mass index; CKD, chronic kidney disease; COPD, chronic obstructive pulmonary disease; IHD, ischaemic heart disease; PAD, peripheral arterial disease; SBP, systolic blood pressure; TIA, transient ischaemic attack.

to those with lower numbers of comorbidities. On average, patients with ≥5 comorbidities had 13.7 (SD 9.0) previous BP readings in the 5 years before hypertension diagnosis versus 6.3 (SD 5.4) previous BP readings in those with hypertension alone, and these differences attenuated over time. However, additional adjustment of the models for this marker of service use had little impact on the inverse relationship between number of comorbidities and SBP (**Fig 4**).

## BP at diagnosis of hypertension

To determine the contribution of BP at diagnosis on long-term BP values, we extracted the BP at the time of diagnosis and adjusted for this value as an additional variable. Overall, this increased the adjusted SBP for those with more comorbidities and decreased it for those with less comorbidities. Indeed, no other covariate had a similarly large effect on follow-up SBP

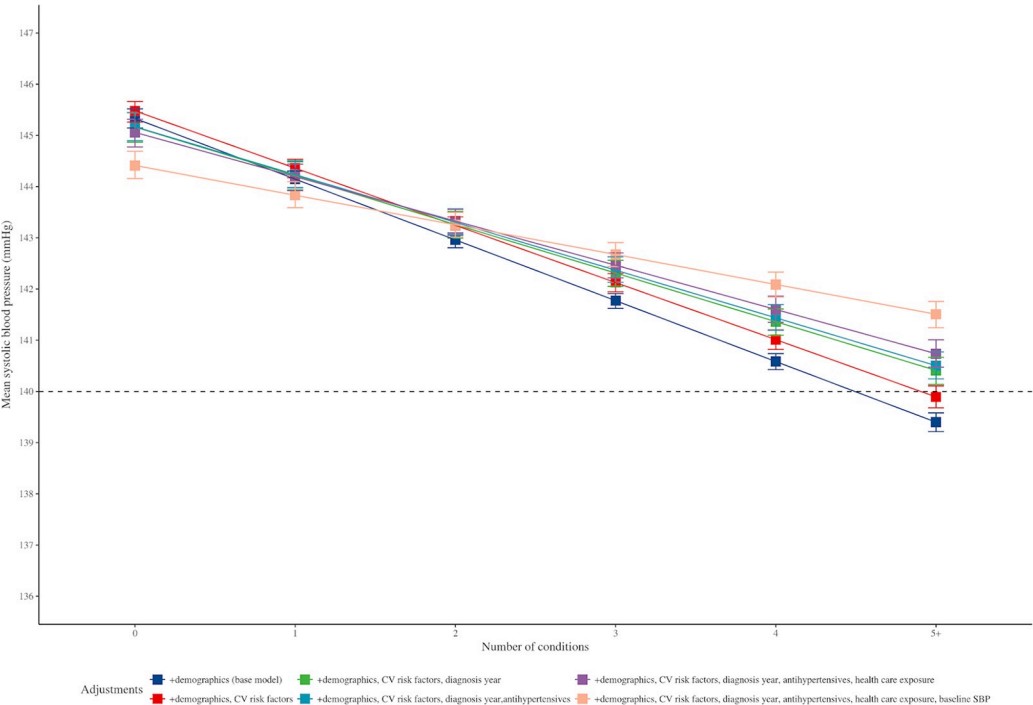

**Fig 4. Association of SBP with number of conditions at 1 year after diagnosis of hypertension, after sequential adjustment of potential confounders.** CV, cardiovascular; SBP, systolic blood pressure.

than the initial difference SBP at the time of diagnosis. With just the adjustment for this covariate, the estimated SBP at 1 year after diagnosis of hypertension in those with ≥5 comorbidities was 141.7 mm Hg (**Fig 4**).

## Discussion

In this large-scale cohort study of patients with newly diagnosed hypertension, we observed a pattern of accumulating comorbidities over time. The presence of higher numbers of comorbidities was associated with lower BP at any time before and after diagnosis of hypertension. Although differences in BP varied by type of comorbidity, we found no evidence that a particular type of comorbidity was associated with a higher BP during follow-up. Previous suggestions of a more rapid decline in BP in those with more comorbidities or particular types of comorbidities were not substantiated during a 10-year follow-up after hypertension diagnosis.

Studies have reported lower BP when an individual has multiple or specific types of comorbidities [4,5]. Some have attributed this pattern to specific biological mechanisms. For example, in patients with dementia, lower BP trajectory was thought to be caused by physical changes (e.g., weight loss and lower total cholesterol), neurodegeneration, and the effect on brainstem regulating centres [6]. Another study in people with rheumatoid arthritis showed an inverse relationship between C-reactive protein and SBP, partly explained by endothelial dysfunction that reduces vasomotor control and BP regulation [24].

Although we are unable to establish the importance of these potential biological disease–disease interactions, we could identify other factors that might explain the lower BP observed in people with more comorbidities and provide quantitative estimates of their impact on BP. One important factor was the higher proportions of antihypertensive prescription in those with particular types of comorbidities. More specifically, we found that in patients with

preexisting cardiometabolic comorbidities, such as ischaemic heart disease, atrial fibrillation, and diabetes, their lower BP was partly explained by a higher proportion of antihypertensive medication use. Adjustment for this treatment slightly attenuated the difference in SBP between those with and without cardiometabolic comorbidity, but BP levels remained lower in patients with other comorbidities despite similar intensity of antihypertensive treatment. Furthermore, the inverse association between the number of comorbidities and SBP persisted even when differences in treatment intensity were adjusted for.

By retrospectively and prospectively determining SBP levels from the time of the diagnosis of hypertension, we found that in people with more comorbidities, SBP remained lower at all times and even before the diagnosis of hypertension compared with those with no additional comorbidity. The initial differences in BP levels persisted for many years after diagnosis for all comorbidity groups, with no evidence of a steeper BP decline among those with more comorbidities. Thus, it seems that the initial bias in diagnosing hypertension at different BP levels is a key determinant of BP differences observed in later years, and the bias was partly attributable to higher frequency of doctor visits and BP measurements in those with more comorbidities.

However, such factors relating to health service interactions could not fully explain the reasons for the lower BP at diagnosis in patients with comorbidities. This finding suggests that perhaps more complex clinical decision-making processes, which may not be captured sufficiently by the indicators we used, determine the timing and threshold of hypertension diagnosis [25]. For instance, the perceived risk of future cardiovascular events or mortality [26] or the potential to benefit from antihypertensive treatment [27] could impact physician and patient preferences for diagnosis and management. Suggestive evidence for this hypothesis came from our exploratory analyses where we found people with cardiovascular multi-morbidity to also have lower cholesterol levels.

Regardless of the underlying causes, the finding of the differential BP levels at which hypertension was diagnosed and its impact on long-term BP trajectories could have implications on making inferences about BP differences between patients with differing numbers or types of comorbidities. For instance, comparing BP levels or "control" rates of hypertension within and between populations [4,28] could lead to erroneous conclusions if BP differences among sub-populations at incident diagnosis are not considered or adjusted for, since even small differences in BP levels at the time of diagnosis, as our data suggest, have substantial and long-term influence on BP differences over time.

The key strength of our study was its longitudinal design and use of large-scale dataset. The CPRD contains information from a large representative sample of the UK population, meaning that our results are generalisable. By leveraging temporal information, we were able select incident hypertensive cases for better comparisons, time-update variables to accurately capture the burden of multi-morbidity, and delineate periods for exposure and outcome variables to establish the temporal sequence of associations and minimise reverse causation. The completeness of prescription data and repeated BP measurements in routine practice add further to the strengths of our study. Moreover, examining BP as a continuous outcome accounts for the full spectrum of risk, as opposed to a binary classification which can lead to loss of information.

However, using practice-setting clinical records has its limitations, including the dependence on recorded diagnoses, which risks underestimating diagnoses due to a lack of inclusion of undiagnosed cases and other practitioner factors. Nonetheless, the reported average positive predictive value and completeness of diagnoses in the CPRD were relatively high when compared to national registries [29]. This is further improved for conditions covered under the UK's QOF incentive scheme, which includes hypertension and the majority of the comorbidities used in our study, that benefit from high reliability of recorded diagnoses [30]. While this approach is likely to underestimate incipient or asymptomatic disease burden, the recorded

diagnosis might indeed be more relevant to decision-making as such data are used to inform service planning and payment. Another limitation is the level of missingness in electronic health record data, which we addressed using sensitivity analyses to determine the best imputation methods to minimise the impact of missing data. Clinical measurements in routine practice are subject to greater measurement errors than epidemiological studies. To mitigate this risk, we modelled BP using multiple measurements, which have been shown to lead to more robust and stronger associations than single BP measurements [31,32]. Nonetheless, this strategy might not have fully addressed this issue, although we find it reassuring that a recent analysis of the Framingham study has also shown a rapidly rising BP in years preceding hypertension diagnosis [33]. Previous studies have shown a strong association between multi-morbidity and frailty [34] with others showing that BP is lower in those with higher frailty [35], thus, raising the hypothesis that similar patterns might be present if frailty is used as the main exposure variable. Future studies could explore this question further.

To our knowledge, no other study has examined BP trajectories longitudinally after diagnosis of hypertension. The fact that BP values several years after diagnosis of hypertension are still sensitive to initial BP at diagnosis suggests that our interpretation of the treatment success might be misleading when such early differences are not considered. Further research could investigate the impact of this "diagnosis bias" on treatment effects for other diseases, like diabetes, which are subject to similar issues. On the other hand, the lack of a more rapid decline in BP in those with multi-morbidity could provide some reassurance for making decision on BP treatment in these high-risk individuals.

In conclusion, our study suggests that the BP at which patients are diagnosed with hypertension is lower when patients have multi-morbidity, and this initial BP difference persists for several years after diagnosis. Despite substantial declines in BP in the first year after diagnosis, subsequent changes were modest, with no evidence of a more rapid decline in those with more or specific types of comorbidities.

## Supporting information

**S1 STROBE Checklist. STROBE, STrenghtening the Reporting of Observational studies in Epidemiology.**
(PDF)

**S1 Fig. Study design schema.**
(PDF)

**S2 Fig. Adjusted mean SBP over time stratified by number of comorbidities. SBP, systolic blood pressure.**
(PDF)

**S3 Fig. Mean SBP over time by number of comorbidities in men. SBP, systolic blood pressure.**
(PDF)

**S4 Fig. Mean SBP over time by number of comorbidities in women. SBP, systolic blood pressure.**
(PDF)

**S5 Fig. Adjusted mean differences in SBP at 1 year after hypertension diagnosis in men, stratified by comorbidity status. SBP, systolic blood pressure.**
(PDF)

**S6 Fig. Adjusted mean differences in SBP at 1 year after hypertension diagnosis in women, stratified by comorbidity status. SBP, systolic blood pressure.**
(PDF)

**S7 Fig. SBP before and at time of hypertension diagnosis, stratified by comorbidities. SBP, systolic blood pressure.**
(PDF)

**S1 Table. Diagnostic codes for hypertension.**
(PDF)

**S2 Table. Comorbidities selected in the study.**
(PDF)

**S3 Table. Antihypertensive British National Formulary codes and drug classes.**
(PDF)

**S4 Table. Descriptive statistics of BP management before and after diagnosis of hypertension. BP, blood pressure.**
(PDF)

**S5 Table. Patient characteristics in landmark cohorts of incident hypertensive patients between 2000 and 2014 in the UK.**
(PDF)

**S6 Table. Baseline characteristics by cardiometabolic comorbidity.**
(PDF)

## Author Contributions

**Conceptualization:** Jenny Tran, Robyn Norton, Kazem Rahimi.

**Data curation:** Jenny Tran, Jose Roberto Ayala Solares, Nathalie Conrad, Francesca Raimondi, Gholamreza Salimi-Khorshidi, Kazem Rahimi.

**Formal analysis:** Jenny Tran, Jose Roberto Ayala Solares, Nathalie Conrad, Francesca Raimondi, Gholamreza Salimi-Khorshidi.

**Funding acquisition:** Jenny Tran, Robyn Norton, Kazem Rahimi.

**Investigation:** Jenny Tran, Robyn Norton, Dexter Canoy, Nathalie Conrad, Gholamreza Salimi-Khorshidi, Kazem Rahimi.

**Methodology:** Jenny Tran, Robyn Norton, Jose Roberto Ayala Solares, Nathalie Conrad, Milad Nazarzadeh, Francesca Raimondi, Gholamreza Salimi-Khorshidi, Anthony Rodgers, Kazem Rahimi.

**Project administration:** Jenny Tran, Robyn Norton, Kazem Rahimi.

**Resources:** Robyn Norton, Kazem Rahimi.

**Software:** Jose Roberto Ayala Solares, Nathalie Conrad, Kazem Rahimi.

**Supervision:** Robyn Norton, Dexter Canoy, Gholamreza Salimi-Khorshidi, Anthony Rodgers, Kazem Rahimi.

**Validation:** Jenny Tran.

**Visualization:** Jenny Tran.

Writing – **original draft:** Jenny Tran.

Writing – **review & editing:** Jenny Tran, Robyn Norton, Dexter Canoy, Jose Roberto Ayala Solares, Milad Nazarzadeh, Anthony Rodgers, Kazem Rahimi.

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
