## [Editor Report · Decision Letter 0]

16 Jul 2020

Dear Dr Rahimi, 

Thank you for submitting your manuscript entitled "Association between comorbidities and blood pressure trajectories in patients with hypertension" for consideration by PLOS Medicine.

Your manuscript has now been evaluated by the PLOS Medicine editorial staff [as well as by an academic editor with relevant expertise] and I am writing to let you know that we would like to send your submission out for external peer review.

Kind regards,

Adya Misra, PhD,

Senior Editor

PLOS Medicine

---

## [Decision Letter · Decision Letter 1]

18 Aug 2020

Dear Dr. Rahimi,

Thank you very much for submitting your manuscript "Association between comorbidities and blood pressure trajectories in patients with hypertension" (PMEDICINE-D-20-03311R1) for consideration at PLOS Medicine. 

[LINK]

In light of these reviews, I am afraid that we will not be able to accept the manuscript for publication in the journal in its current form, but we would like to consider a revised version that addresses the reviewers' and editors' comments. Obviously we cannot make any decision about publication until we have seen the revised manuscript and your response, and we plan to seek re-review by one or more of the reviewers. 

We expect to receive your revised manuscript by Sep 08 2020 11:59PM. Please email us (plosmedicine@plos.org) if you have any questions or concerns.

We look forward to receiving your revised manuscript. 

Sincerely,

Adya Misra, PhD

Senior Editor 

PLOS Medicine

plosmedicine.org

Title

Please revise your title according to PLOS Medicine's style. Your title must be nondeclarative and not a question. It should begin with main concept if possible. "Effect of" should be used only if causality can be inferred, i.e., for an RCT. Please place the study design ("A randomized controlled trial," "A retrospective study," "A modelling study," etc.) in the subtitle (ie, after a colon).

Competing interests

Please add this statement to the manuscript's Competing Interests: "[Initials] is an Academic Editor on PLOS Medicine's editorial board."

DAS

The Data Availability Statement (DAS) requires revision. For each data source used in your study: 

Abstract

* Please structure your abstract using the PLOS Medicine headings (Background, Methods and Findings, Conclusions).

* Please combine the Methods and Findings sections into one section, “Methods and findings”.

* Please ensure that all numbers presented in the abstract are present and identical to numbers presented in the main manuscript text. * Please include the study design, population and setting, number of participants, years during which the study took place, length of follow up, and main outcome measures. * Please quantify the main results (with 95% CIs and p values). * Please include the important dependent variables that are adjusted for in the analyses.

Conclusions

* Please address the study implications without overreaching what can be concluded from the data; the phrase "In this study, we observed ..." may be useful. * Please interpret the study based on the results presented in the abstract, emphasizing what is new without overstating your conclusions. * Please avoid vague statements such as "these results have major implications for policy/clinical care". Mention only specific implications substantiated by the results.

Please revise “multimorbid patients” to “patients with multi-morbidities” or similar 

We do not require a key messages section. We ask that you include a short, non-technical Author Summary of your research to make findings accessible to a wide audience that includes both scientists and non-scientists. The Author Summary should immediately follow the Abstract in your revised manuscript. This text is subject to editorial change and should be distinct from the scientific abstract. Please see our author guidelines for more information: https://journals.plos.org/plosmedicine/s/revising-your-manuscript#loc-author-summary

Please place reference square brackets after punctuation 

Please revise comorbidities throughout to “co-morbidities”

Introduction

The first few sentences are identical to the start of the abstract, please revise 

Methods

Please briefly describe the landmark cohort design

Please ensure that the study is reported according to the [STROBE] guideline, and include the completed [STROBE or other] checklist as Supporting Information. When completing the checklist, please use section and paragraph numbers, rather than page numbers. Please add the following statement, or similar, to the Methods: "This study is reported as per the Strengthening the Reporting of Observational Studies in Epidemiology (STROBE) guideline (S1 Checklist)."

Did your study have a prospective protocol or analysis plan? Please state this (either way) early in the Methods section.

Results

Please provide exact p-values along with 95%CI 

Please simplify this sentence “Patterns were similar in years before and at hypertension diagnosis across

diseases, but with larger differences in years before hypertension diagnosis than years after

diagnosis (Figure S3)” 

Please revise Antihypertensive to “Anti-hypertensive”

Discussion

Please simplify this sentence “we could identify and quantify the impact of additional reasons for the lower BP in people with more comorbidities”

“such as ischemic heart

disease, atrial fibrillation and diabetes, lower BP was partly driven by a higher rate of

antihypertensive use. However, BP levels were also lower in patients” please replace driven by to “associated with” to avoid overstating findings from an observational study 

Please rephrase this for clarity and accuracy “By tracking patients forwards and backwards in time” 

Bibliography- please use Vancouver style 

Comments from the reviewers:

Reviewer #1: I confine my remarks to statistical aspects of this paper. These were mostly fine, but I have a couple minor issue to resolve before I can recommend publication.

NOTE: Line numbers would have made the review easier

p. 6 Describe "quality data" at least a little. Don't just cite a paper.

p 7 Please descibe what a "multiple landmark cohort design" is. 

p. 8 Give more detail about how the values were used to get mean SBP

 Don't categorize BMI. Categorizing any continuous IV is almost always a mistake., In *Regression Modelling Strategies* Frank Harrell lists 11 problems with this. For BMI in particular, see my post https://medium.com/peter-flom-the-blog/why-bmi-is-a-bad-measure-of-obesity-and-what-is-better-f8a62fc9ca49

p. 11 These effects are pretty small. Are they clinically important? I actually take my BP daily and it varies by more then 5 points from day to day

Peter Flom

Reviewer #2: Men and women have very different distributions of blood pressure and comorbidities. Also, there is a large difference between men and women in blood pressure changes after age 60s. Thus sex-stratified analysis should be presented as the main analysis, not as a sensitivity analysis.

It seems that people who took anti-hypertensive drugs before and after the diagnosis of hypertension were included. Also, the prescription before and after diagnosis is not clearly distinguished. People with other diseases would be more likely to take anti-hypertensive drugs before hypertension diagnosis. It can affect the mean blood pressure levels of different groups. If authors want to see the effects of comorbidities on blood pressure level, people with a history of taking anti-hypertensive drugs should be excluded and analyzed.

Many of the comorbidites, which were included in the study, are also associated with high blood pressure or its risk factors. Such hypertension-related diseases and other diseases should be analyzed separately.

Although the authors have mentioned implications for future research and practice, I am not exactly aware of the clinical significance of the results from this study. Please make clear what this study is trying to evaluate: the effects of comorbidities on blood pressure levels; the effects of comorbidities on the diagnosis of hypertension; or the effects of comorbidities on the management of diagnosed hypertension. Depending on the purpose of the study, the selection criteria and analysis methods may vary.

Reviewer #3: This paper described the association between SBP and comorbidities in participants with hypertension. It is interesting to provide evidence for guiding antihypertension therapy. However, there are some major concerns as explained below. I hope the following comments would be useful for the authors to improve this study.

1. As a real-work study based on the linked electronic health records, I think it's important to provide a flowchart to describe how the authors selected the patients from the study population. This will facilitate readers to interpret the results for the generalization and judge any selection bias existed in the study population.

2. It will be very helpful to provide the number of patients without imputation to actually have BP measurement at each year before or after the hypertension diagnosis. It's suspected to see that even five years before the new diagnosis of hypertension, the mean BP levels in all these patients were above 140. It may be argued that there was missing not at random for this type of data, i.e. their BP may be recorded only when its value is above 140.

3. How did the authors accommodate the within-subject correlations when assessing the association between SBP and comorbidities?

4. The authors stated that newly diagnosed hypertension are likely to accrue additional comorbidities referred in the Key messages. How does the results support this? This study population was old (mean age, 61.6y) and it's not clear the evolution of comorbidity in participants without diagnosed hypertension. Patients without any additional comorbidity were still majority of the population.

5. Where did the authors extract the BP measurements from the EHRs? Will it be different for patients with or without comorbidities? For example, patients without comorbidities may be measured by GP whereas patients with other comorbidities could be obtained from the in-patient data.

6. It is not clear how to calculate SBP as time-updated covariates. If it was stated in Text S1, please quote this appendix in the Method.

7. As mentioned in the limitations, missing data was inevitable in EHR data. It's better to briefly report the missing rate of covariates and missing pattern if rate is high.

[LINK]

---

## [Decision Letter · Decision Letter 2]

25 Nov 2020

Dear Dr. Rahimi,

Thank you very much for submitting your manuscript "Multi-morbidity and blood pressure trajectories in hypertensive patients: a multiple landmark cohort study" (PMEDICINE-D-20-03311R2) for consideration at PLOS Medicine. 

[LINK]

In light of these reviews, I am afraid that we will not be able to accept the manuscript for publication in the journal in its current form, but we would like to consider a revised version that addresses the reviewers' and editors' comments. Obviously we cannot make any decision about publication until we have seen the revised manuscript and your response, and we plan to seek re-review by one or more of the reviewers. 

We expect to receive your revised manuscript by Dec 16 2020 11:59PM. Please email us (plosmedicine@plos.org) if you have any questions or concerns.

We look forward to receiving your revised manuscript. 

Sincerely,

Adya Misra, PhD

Senior Editor 

PLOS Medicine

plosmedicine.org

Comments from the reviewers:

Reviewer #1: The authors have addressed my concerns and I now recommend publication. 

Reviewer #3: This paper described the association between SBP and comorbidities in participants with hypertension. Though the dataset is big which described the association between SBP and comorbidities to guide anti-hypertension therapy, there are some major concerns as explained below:

1. In this revision, the authors added information on actual numbers with and without BP measurements and stated that recording of blood pressure is likely to be biased. Considering the assumption of imputation method (mostly missing at random) required, I'm not convinced that this way to deal with missing data is valid. Furthermore, the conclusion based on this, in my opinion, is not solid. The authors claimed, in fact, they would like to conclude that the association is caused by the recording (and treatment) and diagnosis biases. Then this seems to be a false association between SBP and comorbidities to me. Moreover, how could this conclusion help to guide the clinical practice?

2. The authors stated that "newly diagnosed hypertension are likely to accrue additional comorbidities" in the first sentence of the discussion. They draw this conclusion based on the observation that the proportion of those with ≥5 comorbidities increased at 10-year landmark time point. I don't really agree with this because this could be just age effect. The comparison between patients with and without hypertension is the direct support to this statement where the current study has only patients with hypertension.

[LINK]

---

## [Editor Report · Decision Letter 3]

12 May 2021

Dear Dr. Rahimi,

Thank you very much for re-submitting your manuscript "Multi-morbidity and blood pressure trajectories in hypertensive patients: a multiple landmark cohort study" (PMEDICINE-D-20-03311R3) for consideration at PLOS Medicine. We do apologize for the long delay in sending you a decision. 

I have discussed the paper with editorial colleagues and our academic editor and I am pleased to tell you that, provided the remaining editorial and production issues are dealt with, we expect to be able to accept the paper for publication in the journal.

[LINK]

Please let me know if you have any questions, and we look forward to receiving the revised manuscript shortly.   

Sincerely,

Richard Turner, PhD

rturner@plos.org

Requests from Editors:

Please adapt the relevant sentence in the competing interest statement (submission form) to "KR is a member of PLOS Medicine's Editorial Board", or similar.

At line 24 of your abstract, please quote aggregate demographic details for study participants. 

Please also mention the country and data source.

Please add a new final sentence to the "Methods and findings" subsection of your abstract, which should begin "Study limitations include ..." or similar and should quote 2-3 of the study's main limitations. 

You mention a research protocol at line 132. Please attach the relevant document, in addition to a prespecified analysis plan if available, as an attachment, referred to the in text. Please highlight analyses that were not prespecified. 

Please remove the information on competing interests, funding and data access from the end of the main text. This information will appear in the article metadata, via entries in the submission form.

Please use the abbreviation "PLoS Med." consistently in your reference list. 

Please move figure S2 to the main body of the paper. 

Please adapt the label of the attached STROBE checklist to "S1_STROBE_Checklist" or similar, and refer to the file by this label in your methods section. 

At line 214 you refer to RECORD: did you mean to refer to STROBE? Please feel free to substitute a completed RECORD checklist if not. 

In the checklist, individual items should be referred to by section (e.g., "Methods") and paragraph number not by line or page numbers, as the latter change upon publication. 

***

---

## [Editor Report · Decision Letter 4]

25 May 2021

Dear Dr Rahimi, 

On behalf of my colleagues and the Academic Editor, Dr Basu, I am pleased to inform you that we have agreed to publish your manuscript "Multi-morbidity and blood pressure trajectories in hypertensive patients: a multiple landmark cohort study" (PMEDICINE-D-20-03311R4) in PLOS Medicine.

Prior to final acceptance please address the following issues:

1. Please make that "... the presence" at line 24;

2. At lines 37 & 261, the point estimate "141.0" does not seem to fall within the relevant 95% CI and we ask you to check and amend this;

3. Around line 192, please state that "All analyses were prespecified" or make an alternative statement as appropriate; 

3. Please refer to the STROBE attachment at line 218; and

3. Please correct the typo in reference 18.

PRESS

Sincerely, 

Richard Turner, PhD 

rturner@plos.org